# Determining the Household Consumption Expenditure's Resilience towards Petrol Price, Disposable Income and Exchange Rate Volatilities

Thomas Habanabakize 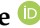

Department of Economic Sciences, North West University, Vanderbjilpark 1900, South Africa; 26767007@nwu.ac.za

**Abstract:** One of the core objectives of economic development is to improve people's standards of living. However, both standards of living and consumption expenditures are often determined by disposable income, crude oil prices and exchange rate volatility. The current paper employed quarterly time series data from 2002 to 2020 to analyse the responsiveness of household consumption expenditure to the petrol price, disposable income and exchange rate volatility in South African. The empirical outcome suggested that a long-run relationship exists between variables under consideration. Additionally, the current level of consumption expenditure was found to be determined by income level and exchange rate volatility whilst changes in petrol price had no significant effect on short-term consumption expenditure. Based on these findings, the study suggests that the South African policymakers and government authorities implement policies and strategies that enhance both household income and exchange rate. Those strategies may include strengthening the country's currency, production improvement, inflation rate reduction, and the creation of job opportunities.

**Keywords:** household consumption; oil price; disposable income; exchange rate; South Africa



## 1. Introduction

The quality of life and consumption expenditure experienced by individuals are often determined by household disposable income, and the latter depends on the country's economic growth. South Africa has been facing slow and volatile economic growth. On average, between 1993 and 2017, the South African annual economic growth was recorded to fluctuate around 2.8 percent. Most of the economic challenges that South Africa undergoes are substantially associated with its linkage to the global markets (Ncanywa and Mgwangqa 2018). For example, the petrol price in South Africa depends on the global market's crude oil price and the strengths or weakness of the South African currency (Rand) against global major currencies such as the US dollar, European Euro, Japanese Yen and the British Pound (Senzangakhona and Choga 2015). Recently both international prices of crude oil and the South African exchange rate have been fluctuating and these fluctuations led to both economic uncertainty and slow economic growth (Balcilar et al. 2017; Nkomo 2006; Patel and Mah 2018). Since the exchange rate and fuel price (petrol price) are linked to the economy and income performance, and the latter is a key driver of consumption expenditure, change in these two economic indicators (petrol price and exchange rate) has significant implications on household consumption expenditure levels.

The relationship between exchange rate and consumption expenditure is often linked to the Backus-Smith puzzle. According to Devereux et al. (2012), the Backus-Smith puzzle refers to the abnormal relationship between consumption expenditure and the real exchange rate. The puzzle is based on the fact that in some cases the consumption expenditure is positively linked to the currency depreciation (exchange rate growth) while in other cases the two economic indicators (the exchange rate and consumption expenditure) are

negatively associated. Besides the effect of currency depreciation on consumption expenditure, the former also influences the petrol price, which in return impacts consumption expenditure (Rangasamy 2017).

The petrol price fluctuation appears to be one of the major issues faced by the South African economy, and its volatility may cause changes in household consumption expenditure. Since the 2008 financial crisis, crude oil prices have experienced significant vacillations. For instance, between 2008 and 2012 the price went up from $40 a barrel to $126 a barrel and declined again to $30.5 barrel in 2016. The price rose again to $86 a barrel in 2018 and declined to $64 a barrel in 2019 (Macrotrends 2020). Owing to the Coronavirus pandemic, in April 2020, the crude oil price was slashed to $18 a barrel (Ambrose 2020) then exceeded $60 in 2021. These fluctuations created economic uncertainty and price instabilities for goods and services. In other words, household consumption expenditure is severely affected by oil price volatility.

Succinctly, changes in petrol price, exchange rate and disposable income influence standard of living and the consumption expenditure. This calls for thorough studies to establish an association amongst the aforementioned economic variables and suggest solutions to the problem. However, to the best knowledge of the author, this gap remains unfilled. Therefore, this paper aims to assess the effects of income, petrol price and exchange rate volatilities on South African household consumption expenditure. The paper is divided into five sections. The first section is the introduction, which provides a concise background; the second section deals with the literature review; the third section discusses data, methodology and approaches employed for empirical analysis; the fourth section provides empirical results and discussion; and the last section, the conclusion, provides concluding remarks and policy recommendations.

## 2. Review of Literature

### 2.1. Consumption and Expenditure Theories

Diverse economists such as Keynes (1936), Duesenberry (1949) and Friedman (1957) were interested in studying household consumption expenditure as one of the key factors that determine a country's wellbeing. From Keynesian consumption expenditure to nowadays, several theories of consumption expenditure's behaviour have been developed (Supriya 2015). These theories include the Consumption Expenditure Theory proposed by Duesenberry in 1949. This theory suggests that the consumption expenditure level does not depend on absolute income but rather on an individual's income. The theory is sometimes alluded to as the relative income theory of consumption expenditure (Ahuja 2013).

Another theory of consumption expenditure was proposed by Modigliani and is referred to as the "Life Cycle Theory of Consumption Expenditure". This theory advocates that the profile of consumption expenditure planned by individuals depends on their lifetime income expectations rather than current disposable income (Gali 1994). Adding to the Modigliani theory of consumption expenditure, Friedman introduced the Theory of Consumer Behaviour. The latter theory is also known as the Permanent Income Hypothesis suggesting that individuals' consumption expenditure does not depend on their present income rather on their permanent income (Supriya 2015). For a better understanding, each of the aforementioned theories is discussed in the subsequent paragraphs.

The Keynesian Absolute Income Hypothesis refers to the social consumption expenditure and is determined by copious factors, be they objective or subjective. Nonetheless, all these factors consider the current level of income as the core determinant of total consumption expenditure (Ezeji and Ajudua 2015). In other words, the Keynesian theory of consumption expenditure asserts that the absolute level of income determines individual and societal consumption expenditure. Consequently, the theory is recognised as the Absolute Income Theory (Alimi 2013). This theory is also linked to the Keynesian psychological law of consumption expenditure stressing that, though at a different pace, a rise in income level results in consumption expenditure growth (Jhingan 2002).

In support of the Keynesian theory, the Relative Income Theory of consumption expenditure argues that an individual's consumption expenditure depends on personal income and its relation to social income. Consequently, if personal absolute income increases while other individuals within the same society do not increase their income with the same percentage, irrespective of his absolute income growth, the individual's consumption expenditure will remain constant (Ohale and Onyama 2002; Supriya 2015). This implies that the variability of individuals' propensity to consume depends on relative income rather than absolute income. Demonstration and ratchet effects are the twin effects that result from absolute income. The demonstration effect postulates that the households' individual consumption expenditure levels may be influenced by the consumption expenditure levels of their families and/or neighbours from the same community (Ohale and Onyama 2002). Conversely, the ratchet effect hypothesises that irrespective of economic situation and decline of income levels, individuals tend to uphold their consumption expenditure level. Further, the ratchet effect posits that during difficult times involving a reduction in income, individuals decide to spend or/and borrow to maintain their consumption expenditure level (Supriya 2015). Following these arguments regarding the Income Theory of Consumption expenditure, one can conclude that, according to this theory, there is no irrefutable relationship between consumption expenditure and income.

In 1954, Modigliani and Brumberg suggested another consumption expenditure theory known as the Life Cycle Theory of Consumption expenditure. This theory argues that an individual's consumption expenditure does not depend on their current income but rather on their lifetime expected income (Gali 1994). Thus, if a person expects to acquire high income in his/her entire life, his/her consumption expenditure level will increase, whilst the consumption expenditure will remain at a low level for a person expecting a modest lifetime income (Deaton 2005). A good example is when a young person inherits his/her parent's wealth. In the beginning, this person expects the income level to remain high; however, as time goes by the reality proves otherwise. Consequently, even when he/she starts working, the level of consumption expenditure diminishes as one portion of income is served for future consumption expenditure. The situation changes at the period of retirement when consumption expenditure increases due to the wealth accumulated during his/her lifetime through savings and/or investment (Gali 1994). Nonetheless, this theory assumes that the person under consideration does not inherit anything from his family yet consumes only his accumulated wealth within the economy where the interest rate and price levels are constant (Ochechuku 1998). Although Deaton (2005) states that the Life Cycle Theory of Consumption expenditure remains relevant, the theory's drawback is manifested in its inelasticity where it ignores the human limitations concerning future emergencies and uncertainties (Supriya 2015). Henceforth, there may exist an inverse relationship between income and consumption expenditure, as individuals save for future consumption expenditure.

In line with the Life Cycle Theory of Consumption expenditure, in 1959 Milton Friedman introduced another theory known as the Permanent Income Theory of Consumption expenditure. The theory also supports the current consumption expenditure and depends on income expected in the long term (Supriya 2015). Friedman (1957) argues that people make decisions on current consumption expenditure based on the income they expect to acquire through wages and return on their investments. Contrary to the Life Cycle Theory of Consumption expenditure, the Permanent Income Theory of Consumption expenditure acknowledges factors that affect wealth accumulation such as exchange rate and interest rates that may also impact on consumption expenditure levels (Supriya 2015).

*2.2. Empirical Evidence on the Linkage between Petrol Price, Disposable Income, Exchange Rate and Consumption Expenditure*

Since factors that affect income and wealth accumulation may impede on consumption expenditure, the next section represents some of the empirical evidence of various factors such as exchange rate and petrol price that influence consumption expenditure.

The exchange rate is one of the macroeconomic variables that may have a significant impact on household consumption expenditure. As illustrated by Ezeji and Ajudua (2015), the exchange rate refers to the value of one currency expressed in terms of other currencies. Thus, when the domestic currency loses its value or rather depreciates, it becomes difficult for domestic consumers to acquire either foreign currencies or goods and services from abroad. Conversely, when the domestic currency appreciates foreign currencies, goods and services become less expensive for domestic consumers. In simple terms, domestic currency depreciation enhances domestic purchasing power (imports become less expensive) whilst their depreciation weakens domestic purchasing power (expensive imports and inexpensive exports). This implies that household consumption expenditure and exchange rate are closely linked (Choi and Devereux 2006). In the United States of America, the validity of this linkage between the aforementioned macroeconomic variables (consumption expenditure and exchange rate) was practically prevalent through a decline of the US dollar for the period ranging between 2000 and 2008 (Heim 2010). During this period, imports become expensive for American consumers. Nonetheless, within the same period, the country (the USA) increased its exports as they were inexpensive within the global markets (Heim 2010). The study of Ezeji and Ajudua (2015) also revealed a positive relationship between consumption expenditure and exchange rate in Nigeria. Appreciation of Nigerian currency (naira) led to high consumption expenditure, while its depreciation caused a decline in consumption expenditure levels. The study of Bahmani-Oskooee et al. (2015) indicated that exchange rate fluctuation results in high inflation, which causes a decline in consumer purchasing power. In support of these findings, the study of Muzindutsi and Thandiwe (2018) confirmed that South African households increase their consumption level as results of Rand appreciation. Thus, an inverse relationship exists between the exchange rate and consumption expenditure. Irrespective of these findings suggesting that currency appreciation and high consumption expenditure level are associated, some other empirical findings (Opazo 2006; Benigno and Thoenissen 2008; Corsetti et al. 2008) attested that consumption expenditure increases when the domestic currency depreciates. Grounded on these findings dichotomy, it can be concluded that the linkage between consumption expenditure level and the exchange rate depends on the country's economic nature and structure.

Irrespective of the high improvement growth of renewable energy sources, fuel energy remains dominant in the energy markets and its shocks affect both countries and individuals' wellbeing (Popp et al. 2014). However, the oil effect on the economy may depend on whether a country is an oil producer (exporter) or oil importer. The petrol price depends also on economic structures and monetary policies adopted by the country (Wang 2013).

A high oil price may lead to high national income and economic growth, thus increasing consumption expenditure, while a high oil price may lead to the high cost of production and thus increase prices for production and reduces consumption expenditure (Ghalayini 2011). In other words, other things being constant, an increase in fuel price generates GDP growth for fuel producer countries while a large portion is spent on the purchase of fuel products for fuel importing countries (Aucott and Hall 2014).

Petrol price has a significant impact on household consumption expenditure. The study of Valadkhani and Mitchell (2002), Kinni (2006) and Rangasamy (2017) found that an increase in petrol price impacts on household and personal consumption expenditure through inflationary growth. An increase in petrol price causes price increases even for non-petrol products. For example, if the cost of transport increases as a result of the petrol price, the cost of production will also increase and, consequently, the selling price for the final products will also follow the same pattern in price growth. Contrary to these findings, the study of Algaeed (2017) revealed that oil price shocks generate income to households in Saud Arabia and consequently increase consumption expenditure.

Besides the petrol price and exchange rate, the consumption rate is also influenced by income levels. Studies conducted by Diacon and Maha (2015) and Keho (2019) in Cote

d'Ivoire, and Lira (2016) in Lesotho indicated that a low level of income restrains private consumption whilst a high level of income results in high households' consumption.

Employment is the source of income for most households. Any factor that reduces employment also has a negative effect on household consumption expenditure. That is, if fuel price affects economic growth (GDP), it also affects employment and thus impacts on consumption expenditure. The study conducted by Alkhateeb et al. (2017) in Saudi Arabia indicated that a positive relationship exists between oil price and employment rate. These results make sense as the country is a producer and exporter of oil. An increase in oil price leads to an increase in both exports and GDP and thus to labour demand. Contrary to Alkhateeb et al.'s (2017) findings, the study Sköld (2020) conducted on the economies of Sweden, Norway, Denmark and Finland showed that an increase in oil prices causes unemployment growth and decline of employment. In support of these contradictory results, the study of Kisswani and Kisswani (2019) in the United States revealed the existence of an asymmetric relationship between oil price and employment levels. Based on these reviewed empirical studies, one can conclude that the effect of petrol (oil) price on employment differs from country to country depending on whether the country is a producer or a consumer of oil products. This conclusion is in line with the finding of Negi (2015) in his study on the effect of oil price on countries' GDP; he found that increase in oil price positively correlated with GDP and employment in Russia and Brazil with negative effects on GDP in China and India.

## 3. Data, Methodological Approach and Model Specification

### 3.1. Description of Data

To analyse the effect of both petrol price and exchange rate volatility on household expenditure, the study adopts the neoclassical production function where household consumption expenditure is expressed by household expenditure at a constant price and petrol price and exchange rate volatility are expressed by their exact values. Quarterly data from 2008Q1 to 2020Q2 is used to determine both long run and short run relationship between the dependent and explanatory variables. The sample period starts in 2008 because the researcher intended to consider the effect of the 2008 financial crisis on household purchasing power. The ending date was selected based on the availability of data. Data for all variables were acquired from Quantec Easy data. The linear function between variables is expressed in Equation (1) as follows:

$$\text{Household expenditure}_t = f(\text{Petrol price, Exchange rate}) \tag{1}$$

The function in Equation (1) can also be expressed in the format of econometric log-linear as follows:

$$LHEXP_t = \beta_0 + \beta_1 LEXP_t + \beta_2 LEXR_t + \beta_3 LINC_t \cdot \varepsilon_t \tag{2}$$

where:

$LHEXP$: is the natural logarithm of total household expenditure
$LPPR$: is the natural logarithm of total petrol price
$LEXR$: is the natural logarithm of the exchange rate
$LINC$: is the natural logarithm of disposable income for a household
$\beta_0$: is intercept (constant) term
$\beta_1$, $\beta_2$ and $\beta_3$: are coefficients of independent variables
$\varepsilon$: is the stochastic error term
$t$: is the period.

### 3.1.1. Approaches

Different econometric statistical approaches and tests are relevant when determining the relationship among variables. These approaches include descriptive statistics, cor-

relation, unit root, cointegration, causality and diagnostic tests. The subsequent section discusses some of these mentioned approaches.

- *Descriptive statistics and correlation*

In quantitative analysis, descriptive statistics plays a vital role in providing a numerical summary of data and measures applied to a specific study (Basawa 2014). Descriptive statistics, specifically, serves as a tool to describe data sample behaviour such as central tendency (mean, median and mode) and dispersion (range, variance, standard deviation and skewness). Based on these measures, descriptive statistics can also be used to compare the study variables. Decent descriptive statistics can reduce the probability of reporting distorted results from research analysis (Plonsky 2015).

- *Correlation*

Correlation analysis is used to measure the extent to which two sample data are statistically related (Sedgwick 2012). The most employed correlation is the Pearson correlation, expressed as follows:

$$r_{xy} = \frac{\sum_{i=1}^{n}(x_i - \overline{x}) - (y_i - \overline{y})}{\sqrt{\sum_{i=1}^{n}(x_i - \overline{x})}\ \sqrt{\sum_{i=1}^{n}(y_i - \overline{y})}} \tag{3}$$

where

$$\overline{x} = \frac{1}{n}\sum_{i=1}^{n} x_i \tag{4}$$

$$\overline{y} = \frac{1}{n}\sum_{i=1}^{n} y_i \tag{5}$$

In Equations (3)–(5) above, $n$ denotes the number of measure value for each variable or set of data. A strong correlation exists between each independent and the dependent variable if $|r_{xy}| \geq 0.5$. For the current study, following the Pearson correlation, the correlation coefficients were calculated to determine the direction and degree to which the independent variables (disposables household income, exchange rate and petrol price) are interrelated with the dependent variable (household expenditure).

- *Stationarity and unit root test*

A time series is considered to be non-stationary if it has a unit root. Thus, its variance depends on time variations and its theoretical correlogram has an inverse relationship with lag length. In other words, as the lag length increases the correlogram diminishes or dies away (Enders 2010). With the presence of a unit rot within a time series, the standard assumptions for asymptotic analysis become invalid as the t-ration does no longer follow a t-distribution. It is, therefore, important to perform a unit root or stationarity test when analysing the time series as the use of nonstationary series may lead to false results and spurious conclusion. The literature encompasses various methods to test for stationarity and/or unit root; the most popular are the Augmented Dickey-Fuller (ADF) test; Phillips-Perron (PP) test; and the Kwiatkowski, Phillips, Schmidt and Shin (KPSS) test (Shrestha and Bhatta 2018). The ADF unit root test is mathematically expressed as follows:

$$\Delta y_t = \mu + \delta_{t-1} + \sum_{i=1}^{k} \beta_i \Delta y_{t-i} + e_t \tag{6}$$

where:
$\Delta y_t$ = the first difference of $y_t$ that is $y_t - y_{t-1}$
$\delta = \alpha - 1$
$\alpha$ = coefficient of $y_{t-1}$

The ADF null hypothesis suggests that $\delta = 0$, and the alternative hypothesis suggests the $\delta < 0$; the null hypothesis is rejected if the series is stationary, and the alternative hypothesis prevails if the series has a unit root. The PP test is the alternative to the ADF test and it is expressed as follows:

$$\Delta y_t = \delta y_{t-1} + \beta D_{t-i} + e_t \tag{7}$$

where

$D_{t-i}$ is a component of a deterministic trend
$e_t$ is a I(0) with zero mean

The ADF and PP test results are almost the same with a slight distinction. The distinction between these two tests is that the PP test does not need to identify the serial correlation of the $\Delta y_t$; in other words, the Phillips-Perron test is non-parametric (Shrestha and Bhatta 2018). Although the same literature opts for ADF to be a more reliable test than the PP test, both tests are considered to be less useful owing to their low power and size distortion (Maddala and Kim 2003).

The abovementioned and other approaches and their classical tests are sometimes biased towards rejection or failure to reject the null hypothesis. Therefore, Kwiatkowski, Phillips, Schmidt and Shin (KPSS) introduced an alternative method that is useful to test for series stationarity. Contrary to the ADF and PP null hypothesis that suggests the presence of a unit root with a series, the KPSS null hypothesis suggests that the series under consideration is stationary while the alternative suggests that it is a non-stationary. The KPSS test is mathematically expressed as follows:

$$Y_t = X_t + \varepsilon_t \text{ and hence } X_t = X_{t-1} + u_t \tag{8}$$

In Equation (8), the hypothesis is tested for $u_t$ and the observed critical value is derived from the LM test statistics. To ensure sound results, all these three tests are performed and compared in this study.

### 3.1.2. ARDL Model Specification

The study employed the autoregressive distributed lag (ARDL) model and bounds testing process to establish the relationship that exists between household expenditure, household disposable income, real exchange rate and petrol price. The following are the advantages that led to the choice of the ARDL model for the current study: as highlighted by Pesaran et al. (2001), the model can consider a different optimal number of lags to seizure series generating procedure from general-to-specific modelling (Laurenceson and Chai 2003). Contrary to traditional models such as Engle and Granger, and Johansen and Jesulius, that—to produce valid results—require large samples and same integration order for variables, the ARDL model produces robust and valid results even when applied on a small sample, and mixed variables' integration provided that none of them are I(2) (Ozturk and Acaravci 2010). Through a simple linear transformation, the error correction model (ECM) is delivered from the ARDL model; and through the ECM short-run, changes are unified with long-run equilibrium. Additionally, the ARDL model allows for the use of distinctive optimum lags within the model (Nkoro and Uko 2016). Furthermore, the ARDL model can be applied to variables with a different order of integration, and it is an adequate model to analyse long-run relationship using a single equation. The estimated ARDL model, for the current study, is derived from Equation (1) and expressed as follows:

$$\Delta LHEXP_t = \beta_0 + \sum_{i=1}^{p} \beta_{1i}\Delta LHEXP_{t-1} + \sum_{i=0}^{q} \gamma_i \Delta LINC_{t-i} + \sum_{i=0}^{m} \delta_i \Delta LEXR_{t-i} + \sum_{i=0}^{n} \theta_i \Delta PPR_{t-i} + \\ \varphi_1 LHEXP_{t-1} + \varphi_2 LINC_{t-1} + \varphi_3 LEXR_{t-1} + \varphi_4 LPPR_{t-1} + \varepsilon_{1t} \tag{9}$$

where $\beta_1$, $\gamma_i$, $\delta_i$ and $\theta_i$ are short-run coefficients and $\varphi_1$ to $\varphi_4$ are long-run coefficients. The null hypothesis of no-cointegration, $H_0$: $\varphi_1 = \varphi_2 = \varphi_3 = \varphi_4 = 0$, is tested against the alternative $H_1$: $\varphi_1 \neq \varphi_2 \neq \varphi_3 \neq \varphi_4 \neq 0$ and the decision is made based on the comparison between the Pesaran et al. (2001) tabulated critical values and the F-test results. A long-run relationship exists among variables if the F-value is greater than the selected upper bound critical value. However, if the calculated *F*-value lies between lower and upper bound critical values, then the results are inconclusive.

For the short-run dynamics, the ECM is derived from the ARDL model and is expressed as follows:

$$\Delta LHEXP_t = \beta_0 + \sum_{i=1}^{p} \beta_{1i}\Delta LHEXP_{t-1} + \sum_{i=0}^{q} \gamma_i \Delta LINC_{t-i} + \sum_{i=0}^{m} \delta_i \Delta LEXR_{t-i} + \sum_{i=0}^{n} \gamma_i \Delta PPR_{t-i} + \omega ECT_{t-1} + u_0 \tag{10}$$

where $\omega$ is the coefficient of the error correction term.

The reviewed literature in Sections 2.1 and 2.2 suggested causation between household consumption expenditure, exchange rate, income and petrol price. Similarly, if two or more variables have a long-run relationship or cointegrate, there should be a causal relationship be it unidirectional or bidirectional. Therefore, if the ARDL model establishes a long-run relationship among variables then the causality estimation is conducted. Nonetheless, the ordinal Granger causality test introduced by Granger (1969) requires series to integrate at the same order and yet the KPSS results, in this study, provided a mixture of I(0) and I(1). To achieve valid causality results from series with different integration orders, a new approach of Granger causality known as the Toda and Yamamoto test for causality has to be employed (Mavrotas and Kelly 2001; Toda and Yamamoto 1995). The following equations were established to estimate the Toda and Yamamoto (T–Y) test:

$$LHEXP_t = \alpha_1 \sum_{j=1}^{k} \beta_{1j}LHEXP_{t-j} + \sum_{i=k+1}^{k+dmax} \beta_{1i}LHEXP_{t-i} + \sum_{j=1}^{k} \gamma_{1j}LINC_{t-j} + \sum_{i=k+1}^{k+dmax} \gamma_{1i}LINC_{t-i} + \sum_{j=1}^{k} \delta_{1j}LEXR_{t-j} + \sum_{i=k+1}^{k+dmax} \delta_{1i}LEXR_{t-i} + \sum_{j=1}^{k} \varphi_{1j}PPR_{t-j} + \sum_{i=k+1}^{k+dmax} \varphi_{1i}PPR_{t-i} + e_{1t} \tag{11}$$

$$LINC_t = \alpha_2 + \sum_{j=1}^{k} \beta_{2j}LHEXP_{t-j} + \sum_{i=k+1}^{k+dmax} \beta_{2i}LHEXP_{t-i} + \sum_{j=1}^{k} \gamma_{2j}LINC_{t-j} + \sum_{i=k+1}^{k+dmax} \gamma_{2i}LINC_{t-i} + \sum_{j=1}^{k} \delta_{2j}LEXR_{t-j} + \sum_{i=k+1}^{k+dmax} \delta_{2i}LEXR_{t-i} + \sum_{j=1}^{k} \varphi_{2j}PPR_{t-j} + \sum_{i=k+1}^{k+dmax} \varphi_{2i}PPR_{t-i} + e_{2t} \tag{12}$$

$$LEXR_t = \alpha_3 + \sum_{j=1}^{k} \beta_{3j}LHEXP_{t-j} + \sum_{i=k+1}^{k+dmax} \beta_{3i}LHEXP_{t-i} + \sum_{j=1}^{k} \gamma_{3j}LINC_{t-j} + \sum_{i=k+1}^{k+dmax} \gamma_{3i}LINC_{t-i} + \sum_{j=1}^{k} \delta_{3j}LEXR_{t-j} + \sum_{i=k+1}^{k+dmax} \delta_{3i}LEXR_{t-i} + \sum_{j=1}^{k} \varphi_{3j}PPR_{t-j} + \sum_{i=k+1}^{k+dmax} \varphi_{3i}PPR_{t-i} + e_{3t} \tag{13}$$

$$LPPR_t = \alpha_4 + \sum_{j=1}^{k} \beta_{4j}LHEXP_{t-j} + \sum_{i=k+1}^{k+dmax} \beta_{4i}LHEXP_{t-i} + \sum_{j=1}^{k} \gamma_{4j}LINC_{t-j} + \sum_{i=k+1}^{k+dmax} \gamma_{4i}LINC_{t-i} + \sum_{j=1}^{k} \delta_{4j}LEXR_{t-j} + \sum_{i=k+1}^{k+dmax} \delta_{4i}LEXR_{t-i} + \sum_{j=1}^{k} \varphi_{4j}PPR_{t-j} + \sum_{i=k+1}^{k+dmax} \varphi_{4i}PPR_{t-i} + e_{4t} \tag{14}$$

where dmax indicates the maximal integration order for all series. In the above models, the null hypothesis (H_0) suggests that the lagged value (coefficient) of each series (variable) is zero, meaning the absence of Granger causality between variables; the alternative hypothesis (H_A) suggests the presence of causality. These hypotheses were assessed using both unrestricted regression and the Modified Wald (MWALD) test. Additionally, different diagnostic tests were performed to ensure the validity of ARDL and T-Y models.

## 4. Empirical Findings and Discussion

### 4.1. Descriptive Statistics and Correlation Analysis

Table 1 represents the summary of information on descriptive statistics of variables used in the study. The mean for underpinned variables is 14.395 for LHEX, 4.575 for LEXCR, 14.582 for LINC and 7.0297 for LPPR, respectively. The LINC has a high mean, while the LHEX has the lowest mean. This implies that the quarterly household disposable income is of a large magnitude and the exchange rate is of the lowest magnitude in comparison to other variables. Taking into account the variability of each variable for the analysed period as represented by the standard deviation, household expenditure and petrol price experienced high volatility compared to exchange rate and household income. All variables except exchange rate are negatively skewed. The skewness value of each variable is close to zero, suggesting that they are normally distributed. The normal distribution suggested by skewness is supported by Jarque-Bera statistics of 4.31 for LHEX, 3.54 for LINC and 4.14 for LPPR. Based on Jarque-Bera values, LEXR is the only variable that is not normally distributed.

### 4.2. Unit Root Test

As mentioned in Section 3.1, unit root tests and stationarity tests were applied to collected data to determine the integration order for each variable. These tests were

conducted to ensure that variables are either integrated at the level, first difference or second difference, as the ARDL model produces defective results if applied on variables that are integrated of the second-order {I(2)}. The results from the ADF, PP and KPSS tests are represented in Table 2. Both ADF and PP revealed that all variables are stationary at the first difference. However, the KPSS results indicate that considering trends, the variable is stationary at levels. According to Imam et al. (2016), if the unit root and stationarity test results are conflicting, the KPSS results are preferred over the ADF and PP results. Besides, none of the study variables is I(2); therefore, the ARDL model is the convenient model to assess the long-run relationship among variables.

**Table 1.** Descriptive statistics results.

| | Descriptive Statistics | | | |
|---|---|---|---|---|
| **Variable** | **Mean** | **Standard Deviation** | **Skewness** | **J-B** |
| LHEX | 14.395 | 0.075 | −0.505 | 4.312 |
| LEXR | 4.575 | 0.113 | 0.127 | 0.478 |
| LINC | 14.582 | 0.263 | −0.208 | 3.542 |
| LPPR | 7.0297 | 0.241 | −0.630 | 4.137 |

Note: J-B denotes Jarque-Bera.

**Table 2.** Results for stationarity and unit root tests.

| **Variables** | **Model** | **Levels** | | | **1st Difference** | | |
|---|---|---|---|---|---|---|---|
| | | **ADF** | **PP** | **KPSS** | **ADF** | **PP** | **KPSS** |
| HEX | Constant | 0.880 | 0.860 | 0.886 | 0.010 * | 0.011 * | 0.097 * |
| | Constant and trend | 0.929 | 0.651 | 0.143 * | 0.048 * | 0.049 * | - |
| EXR | Constant | 0.430 | 0.386 | 0.271 ** | 0.000 ** | 0.000 ** | - |
| | Constant and trend | 0.518 | 0.476 | 0.099 * | 0.000 ** | 0.000 ** | - |
| INC | Constant | 0.913 | 0.915 | 0.936 | 0.002 ** | 0.000 ** | 0.159 * |
| | Constant and trend | 0.825 | 0.825 | 0.128* | 0.012 * | 0.000 ** | - |
| PPR | Constant | 0.366 | 0.407 | 0.802 | 0.000 ** | 0.000 ** | 0.162 * |
| | Constant and trend | 0.239 | 0.232 | 0.107 * | 0.000 ** | 0.000 ** | - |

Note: **, * denotes stationarity at 1% and 5% significant levels, respectively. Critical values for KPSS are 0.463 and 0.1460 with intercept and trend, respectively, at a 5% level.

### 4.3. Pairwise Correlation

Since the stationarity of variables is determined, it is now possible to apply the Pairwise correlation test to empirically determine whether two or more variables are correlated or interdependent. The aim of the test is to establish the magnitude and signs of correspondence between the study variables. Results in Table 3 suggest a strong and positive correlation between household consumption expenditure, income and petrol price. However, a weak and negative correlation exists between household consumption expenditure and exchange rate.

**Table 3.** Pairwise correlation results.

| **VARIABLES** | **LHEX** | **LEXR** | **LINC** | **LPPR** |
|---|---|---|---|---|
| LHEX | 1.000 | | | |
| LEXR | −0.394 ** | 1.000 | | |
| LINC | 0.762 *** | −0.360 *** | 1.000 | |
| LPPR | 0.673 *** | −0.387 ** | 0.830 | 1.000 |

Note: ** and *** denote the significance of correlation at 0.05 and 0.01 levels, respectively.

### 4.4. Assessment of the Long-Run Relationship

Using the ACI information criteria, the ARDL (3,4,1,1) appeared to be an appropriate model for the long-run and short-run relationships between variables. The bound test results are displayed in Table 4. These results suggest that the value of the estimated F-statistics is greater than all the upper bound critical values. Consequently, the null hypothesis of no cointegration is rejected in favour of the alternative hypothesis. In other words, a long-run relationship exists between household expenditure, household income, exchange rate and petrol price in the South African economy. The estimated relationship is represented by Equation (11).

$$LHEX = 10.5778 + 0.0102 * LEXR + 0.2108 * LINC + 0.0963 * LPPR \qquad (15)$$

Equation (15) indicates that changes in LEXR, LINC and LPPR have positive long-run impacts on LHEX. The coefficient of LEXR suggests that a one percent increase in the exchange rate, keeping other factors constant, increases the household consumption expenditure by 0.0102 percent. Additionally, a one percent increase in household income is associated with a 0.2108 percent increase in household expenditure, other factors being constant. Furthermore, other factors being held constant, a one percent increase in petrol price leads to a 0.0963 increase in household expenditure. These study findings are in line with Algaeed (2017), Diacon and Maha (2015) and Muzindutsi and Thandiwe (2018), who found a positive long-run effect of the exchange rate, household income and oil price on household expenditure.

**Table 4.** ARDL bounds test for cointegration.

| F-Bounds Test | | Null Hypothesis: No Levels Relationship | | |
|---|---|---|---|---|
| Test Statistic | Value | Sign in. | I(0) | I(1) |
| F-statistic | 10.04703 | 10% | 2.37 | 3.2 |
| k | 3 | 5% | 2.79 | 3.67 |
| | | 1% | 3.65 | 4.66 |

Although South Africa is not a petrol producer, these results suggest that an increase in petrol price does not impede household consumption expenditure provided that the South African currency remains strong and household income is increasing. Currency appreciation and household income improvement are the major strategies that can improve South African household expenditure.

### 4.5. Analysis of the Short-Run Dynamisms

Prior to the analysis of short-run dynamism between variables, diagnostic tests were performed to ensure the accuracy of findings. The error correction model (ECM) has passed all performed diagnostic tests, as indicated by results reported in Section 4.6. The required features of the acceptable error correction term were met (negative and significant with a *t*-value of [7.505]). The ECT coefficient of −0.324 suggests that around 32 percent of shocks in the model (from the equilibrium) are corrected each quarter. That is, fluctuations in the exchange rate, household income and petrol price take approximately 3.089 (1/0.324) quarters to have a significant effect on household consumption expenditure.

Following Equation (10), the short-run effect of regressors on the dependent variable was determined. The results reported in Table 5 suggest that current consumption expenditure can positively be influenced by the consumption expenditure growth of two previous quarters. Additionally, income growth was found to have a positive short-run impact on current consumption expenditure. Short term dynamics of exchange rate and petrol price does not significantly influence household consumption expenditure. In line with long-run results, policymakers must ensure the stability of the exchange rate and enhancement of household income to improve household expenditure.

**Table 5.** ECM and short-run dynamics.

| Variable | Coefficient | Standard Error | t-Statistic | Probability |
|---|---|---|---|---|
| D(LHEX(−1)) | 0.266140 | 0.084986 | 3.131576 | 0.0036 |
| D(LHEX(−2)) | 0.314134 | 0.094211 | 3.334363 | 0.0021 |
| D(LEXR) | −0.001543 | 0.007802 | −0.197748 | 0.8445 |
| D(LINC) | 0.510629 | 0.054263 | 9.410262 | 0.0000 |
| D(LPPR) | 0.003545 | 0.005567 | 0.636738 | 0.5287 |
| ECT | −0.323699 | 0.043131 | −7.504951 | 0.0000 |

Source: own construction.

### 4.6. Analysis of the Causal Relationship

As mentioned in the methodology section, the presence of cointegration among variables implies that there should be at least one causal relationship between household expenditure and its explanatory variables, be it unidirectional or bidirectional. The T–Y approach was applied to equations 11 to 14 to determine the causal relationship between variables. The results exhibited in Table 6 (column2) suggest that growth in household consumption expenditure is Granger-caused by both exchange rate and income growth. These results support those in short-run dynamics (in Table 5), suggesting that both LEXR and LINC significantly influence short-term changes in household consumption growth. In other words, an increase in household income and appreciation of South African currency leads to high consumption growth while a decline in household income and exchange rate causes a decline in household consumption. Results in Table 6, however, suggest a unidirectional causality from the household consumption expenditure to growth in petrol price. On the other hand, the growth in household consumption expenditure can Granger-cause changes in both income and petrol price. Considering its significance at 10 percent, changes in income growth can Granger-cause changes in the growth of consumption expenditure, exchange rate and petrol price. Furthermore, results in Table 6 column 6 suggest that all explanatory variables (LHEX, LEXR, LINC and LPPR) jointly Granger-cause changes in household expenditure's growth. This implies that short-term changes in LHEX can be influenced by growth in LEXR, LINC and LPPR. Given the tie that exists between household expenditure, income, exchange rate and petrol prices, fiscal and monetary policy such as job creation, production growth and inflation reduction are tools that can assist not only in fostering economy but also improving household expenditure and standard of living.

**Table 6.** Causality between household expenditure, exchange rate, income and petrol price.

| Excluded Lags | Dependent Variable | | | | Explanatory Combined |
|---|---|---|---|---|---|
| | LHEX | LEXR | LINC | LPPR | |
| **LHEX** | - | 1.397365 | 4.600898 | 11.58947 | 21.65490 |
| | - | (0.2372) | (0.0312) | (0.0007) | (0.0001) |
| **LEXR** | 20.18872 | - | 2.595125 | 0.774053 | 13.41009 |
| | (0.0000) | - | (0.1072) | (0.3790) | (0.0038) |
| **LINC** | 5.047126 | 3.706724 | - | 3.800898 | 6.799751 |
| | (0.0247) | (0.0542) | - | (0.0512) | (0.0786) |
| **LPPR** | 0.162441 | 2.709002 | 1.073403 | - | 17.30356 |
| | (0.6869) | (0.0998) | (0.3002) | - | (0.0006) |

*p*-values in brackets.

### 4.7. Diagnostic Tests Results

To ensure that the estimated ARDL model met required econometric assumptions, residuals and stability tests were performed. Results in Table 7 indicate that the model passed all conducted tests. The null hypotheses for homoscedasticity, normal distribution and no autocorrelation were not rejected, meaning that the study residuals or error term

were homoscedastic, free of serial correlation and normally distributed. The stability of parameters in the model was also proved by the Ramsey RESET results, and this was also confirmed by CUSUM graphs as displayed in Figure 1 below. Parameter stability infers the relationship that exists between the study variables was consistent during the analysed period.

**Table 7.** Diagnostic results summary.

| Test | Null Hypothesis | P or *F*-Value | Decision |
|---|---|---|---|
| Jarque-Bera (JB) | Residual is multivariate normal | 0.4452 | H0 Not rejected |
| LM Test | No serial correlation | 0.3869 | H0 Not rejected |
| White | No heteroscedasticity | 0.4064 | H0 Not rejected |
| Ramsey RESET Test | The model is properly specified | 0.3517 | H0 Not rejected |

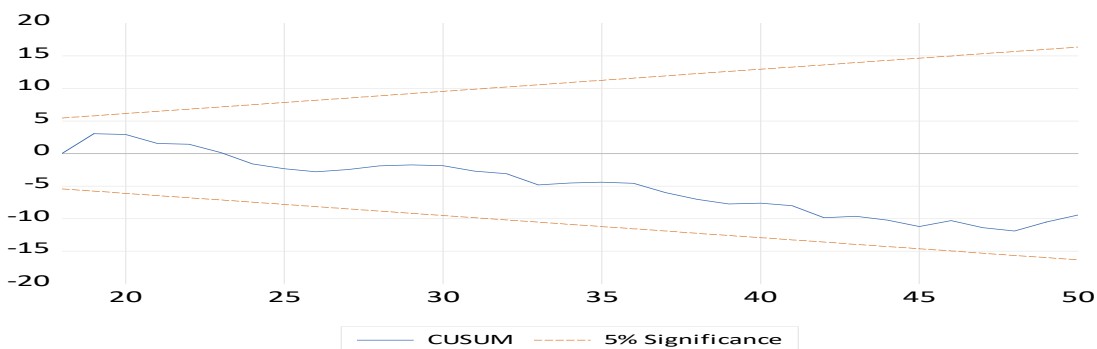

**Figure 1.** CUSUM test results.

## 5. Empirical Significance of the Study and Managerial Implication

The positive relationship found in this study's analysis implies that an increase in household consumption creates a high demand for energy (fuel) to produce what is required by households. These results support Mukhtar et al. (2020) study, whose findings suggested a positive cointegration between demand for energy and household consumption. On the other hand, the positive relationship between exchange rate and household consumption (found in this study's analysis) was similarly obtained by Ezeji and Ajudua (2015) and Muzindutsi and Thandiwe (2018). Exchange rate and fuel price also influence income growth, and the latter is a core factor of consumption growth.

Given that South Africa is an importer of petrol products, the negative impact that exists between petrol price and consumption can be managed through strengthening and stabilizing the exchange rate. With a strong exchange rate, the cost of production within domestic industries becomes low and imports become less expensive, leading to possible growth in consumption expenditure. A steady and strong exchange rate means low inflation and high purchasing power for households. Based on the study findings, consumption expenditure and standard of living in South Africa can be improved if the South African monetary and fiscal authorities ensure the stability of the currency. An increase in the stock of energy and the provision of various means that generate income to households such as job creation and skills development can also increase the consumption expenditure.

## 6. Conclusions

Despite the fact that the literature represents several studies on the effect of the exchange rate, income and petrol price on household consumption expenditure, there is a shortage of studies that analyse the effect of the aforementioned variables in the South African context using a single equation. Therefore, this study assessed the cointegration

and the short-run relationship between household expenditure, household income and petrol price in the South African economy. The study employed the ARDL approach as it is known to be a robust model that uses a single equation and generates accurate results from a mixture of integration order. The findings of this study revealed that a long-run relationship exists between the analysed four variables. All three independent variables were found to have a positive long-run impact on household expenditure. The short-run results from ECM indicated that the current consumption expenditure's growth can be influenced by its own lagged values and the current income growth. The T–Y Granger non-causality results revealed a bidirectional relationship between household income and consumption expenditure. Additionally, household consumption expenditure may cause changes in the exchange rate, household income and petrol price behaviours. Furthermore, it was also found that short-term changes in the exchange rate may be caused by petrol price, household income and household expenditure growth.

The study findings indicate that improving household expenditure and standard of living in South Africa will depend on strengthening the country's currency and household income, and on the availability of natural oil (petrol). Therefore, policies focusing on job creation, production growth, inflation reduction and exchange rate stability would help improve both household expenditure and South African welfare. While the literature argues that the positive relationship between petrol price and consumption expenditure would more likely be experienced in countries that produce oil, the current study results prove otherwise for the South African case. Therefore, future studies should analyse causes of the positive effect of petrol price on the South African household expenditure. A study that uses non-linear approaches would assist in bringing a solution to the positive linkage between consumption growth and oil price.

**Funding:** This research received no external funding.

**Data Availability Statement:** Data available on request due to restrictions (subscription fee). For subscribed individuals or institutions, this data can be found here: (https://www.quantec.co.za/easydata/).

**Conflicts of Interest:** The author declares no conflict of interest.

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
