# Peer review of "Determining the Household Consumption Expenditure’s Resilience towards Petrol Price, Disposable Income and Exchange Rate Volatilities"

_economies, doi:10.3390/economies9020087_

Round 1

Reviewer 1 Report

The aim of the paper is to determine the effect of income, petrol price and the exchange rate volatility on the South African household consumption expenditure. The paper provides some literature review related to this matter, there is however no connection between economic theory and the empirical data analysis used in the paper. It is unclear how this advanced econometric approach stems from  eq. (1) and vice versa, how this econometric theory can contribute to better understanding of eq(1).

Another issue is related to correlation test. Authors present results of correlation analysis on the variables, that are nonstationary. These results are biased and such correlations are proven to be spurious (cf. Kao, 1999).

Author Response

Thanks for the constructive comments.

The attached file represents information on action taken based on the reviewer's report (comments and suggestions). 

Reviewer 2 Report

In this paper, the authors analyze the short and long-run relationship between consumption expenditure, petrol price, disposable income, and exchange rate volatility using quarterly time series from 2002 to 2020 in the South African economy. The estimation results using the ARDL model with the error-correction term and the Granger-causality tests indicate that current consumption expenditure is affected by income and exchange rate volatility while the petrol prices do not have an effect on the expenditure. Overall, I think the manuscript can be improved much better if the authors can address and clarify the following issues.

(Specific comments)

  1. Throughout the manuscript, the expression “current level of consumption” appears to be corrected into “current growth rates of consumption” or similar expressions. In the short-run dynamics shown in Equation (9), the interesting variables are all in the form of log-difference, indicating the growth rates of the variables.
  2. With respect to “2. Review of literature”, I strongly recommend that the authors rewrite it more succinctly by focusing on previous empirical studies exploring the relationship between consumption expenditure and income, exchange rate volatility, and petrol prices.
  3. Relatedly, I found several repeated or redundant sentences or paragraphs in the 2. Review of literature, which should be revised appropriately.
  4. For the bound test for cointegration in line 366, it would be recommendable that the authors add some explanations on the test or relevant references. For example, why the bound test is adopted rather than the conventional Johansen test etc.
  5. In line 387, the previous literature Algaeed (2017), Diacon and Maha (2015), and Muzindutsi and Mjeso (2018) should be addressed in section 2. Review of literature. Also, I am wondering whether the estimated signs of cointegration shown in Equation (11) are overall consistent with those found in the above three studies.
  6. In line 427, I am also wondering whether the Granger-causality test results are consistent with those found in previous literature. Relatedly, I am afraid that there is a little bit of conflict in the main results of the paper in that the lagged growths of income and exchange rates have significant effects on current consumption growth according to Table 4, but the Granger-causality tests indicate consumption growth only causes growths of income, exchange rates, and petrol prices, not the cases of reverse causal flows as illustrated in Table 5. Authors need to reconcile those estimation results.

Author Response

Thanks for constructive comments.

The attachment contains information about action taken based on comment and suggestions from the reviewer.

Round 2

Reviewer 1 Report

Thank you for addressing most of my comments. Can you please clarify what is the purpose of correlation analysis for nonstationary data in your case ?

Author Response

As suggested the correlation analysis was analysed using stationary variables. The results are reported in Table 3, page 10.

Reviewer 2 Report

Even though the literature review section has been improved, the overall questions raised in the first review are less successful to clarify the issues.

(1) Based on the expression in Equations (9) and (10), Table 4 and Table 5, the consumption growth appears to be used, which is also appropriate in the framework of the short-run dynamics. However, the authors keep arguing these variables are consumption level or income level. I am very confused about it. The authors need to further clarify that. As noted in the first review, those variables have a log-difference format. If these variables are log levels as the authors argue, then the short-run dynamics and Granger-causality tests have rarely methodological rationale in that they are all estimated using the OLS upon stationary variables. 

(2) For the bound test for cointegration, I cannot find any appropriate explanations on why the bound test is adopted rather than the conventional Johansen test.

(3) I raised the following issue that the Granger-causality test results are less consistent with those found in Table 4. That is, the lagged income (growth) and exchange rates have significant effects on current consumption (growth) according to Table 4, but the Granger-causality tests indicate consumption growth only causes growths of income, exchange rates, and petrol prices, not the cases of reverse causal flows as illustrated in Table 5. Then, the authors just revised the estimated coefficients from insignificancy to significance without fully specified explanations. 

Author Response

  1. Following equations 9 to 14 of the manuscript, the short-run dynamics were established using Error Correction Model (ECM) and the causation was obtained through Toda-Yamamoto Granger non-causality test. the interpretation of these results took logs (growth) into consideration.
  2. As explained on page 7 of the manuscript, the bound test for cointegration was selected based on the following: (1) traditional models such as Johansen and Engle and Granger employ system equations while ARDL uses single equation (the aim of this study). (2) traditional approaches require variables with same integration order yet based on KPSS, variables of this study are integrated of different order. (3) Bound test allows the used of various number of lags (each variable can have its optimal lags) while traditional apply same number of lags to all variables in the study.....
  3. New analysis for causality were performed and new results are reported and interpreted on page 11 and 12.
  4. All sections of the manuscripts were revised based on provided comments.